# SYNPLAY: IMPORTING REAL-WORLD DIVERSITY FOR A SYNTHETIC HUMAN DATASET

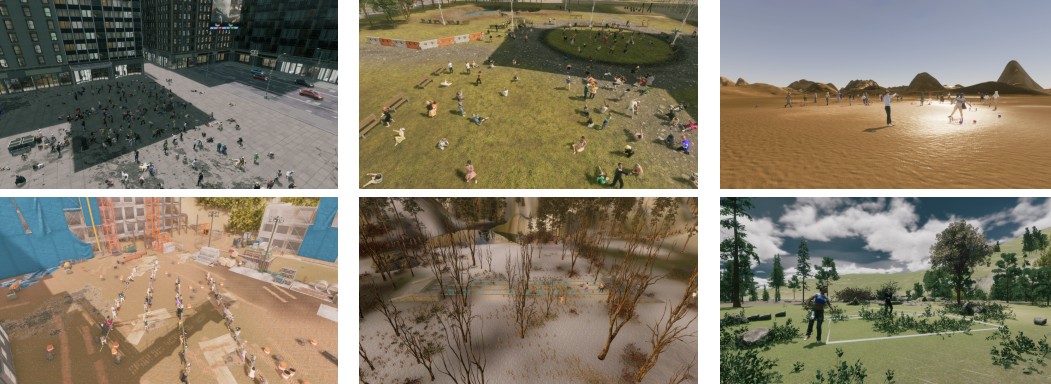

Figure 1: **SynPlay** dataset is constructed while players play six traditional games in a virtual playground also introduced in the Netflix TV show "Squid game" (Hwang, 2021). We have diversified the human appearances in the scenes by focusing on two factors: i) leveraging real-world human motions and ii) adopting multiple viewpoints.

## ABSTRACT

We introduce Synthetic Playground (SynPlay), a new synthetic human dataset that aims to bring out the diversity of human appearance in the real world. We focus on two factors to achieve a level of diversity that has not yet been seen in previous works: i) realistic human motions and poses and ii) multiple camera viewpoints towards human instances. We first use a game engine and its library-provided elementary motions to create games where virtual players can take less-constrained and natural movements while following the game rules (*i.e.*, rule-guided motion design as opposed to detail-guided design). We then augment the elementary motions with real human motions captured with a motion capture device. To render various human appearances in the games from multiple viewpoints, we use seven virtual cameras encompassing the ground and aerial views, capturing abundant aerial-*vs*-ground and dynamic-*vs*-static attributes of the scene. Through extensive and carefully-designed experiments, we show that using SynPlay in model training leads to enhanced accuracy over existing synthetic datasets for human detection and segmentation. The benefit of SynPlay becomes even greater for tasks in the data-scarce regime, such as few-shot and cross-domain learning tasks. These results clearly demonstrate that SynPlay can be used as an essential dataset with rich attributes of complex human appearances and poses suitable for model pretraining. SynPlay dataset comprising over 73k images and 6.5M human instances, will be publicly released upon acceptance of this paper.

## 1 INTRODUCTION

Large-scale synthetic datasets, known for their scalability, provide a practical solution to the increasing demand for training large-capacity models. Recently developed rendering engines (*e.g.*, Unity, Unreal, *etc.*) have significantly enhanced the realism of synthetic data, broadening its applicability

across various computer vision tasks. Despite efforts to scale synthetic data to match the extensive curation of real-world data, the desired level of diversity has not yet been achieved. This insufficiency in diversity is largely due to the inadequate consideration and integration of key factors that are essential to real-world diversity in the process of creating synthetic data.

Recently, several attempts have been made to increase human appearance diversity by controlling innate characteristics (*e.g.*, race, gender) (Black et al., 2023), body shape (*e.g,* height) (Patel et al., 2021; Black et al., 2023), or clothing (Black et al., 2023). These datasets have demonstrated their effectiveness in tasks aimed at identifying human characteristics from close-up images, *e.g.*, human body/pose estimation (Patel et al., 2021; Black et al., 2023) and shape reconstruction (Mahmood et al., 2019). However, these datasets have seldom yielded a discernible positive impact on computer vision tasks aimed at identifying humans *from a distance*, *e.g.,* human detection and segmentation.

When it comes to recognizing the overall human appearance from a distance, the motions and poses exhibited by the individuals play more vital roles than other characteristics. Despite prior attempts to synthesize various human poses, the quality of the rendering remained suboptimal, lacking in realism (Richter et al., 2016) and diversity (Shen et al., 2023b). AMASS (Mahmood et al., 2019) was the output of an early endeavor aimed at achieving both realism and diversity, where a motion scanner was utilized to collect real human motions. Before capturing these motions, detailed descriptions were provided to articulate specific movements — *e.g., 5 seconds waving above the head with both arms*[1], while adhering to physical constraints that limit large movements in motion-capture environments. This *detail-guided motion design* often results in capturing a restricted range of motions tied to specific descriptions while missing out on all the motions that defy easy description.

We claim that providing relatively high-level, less-detailed guidance greatly helps in breaking out from the aforementioned limitations and provides more freedom towards the expansion of the diversity in human motions. In constructing our dataset, we follow the new **rule-guided motion design** approach providing game "rules" or winning strategies for the virtual players to follow, which serve as a set of significantly coarser guidelines when compared to detail-guided approaches. In this way, the motions that they manifest are not confined to predetermined/easily-describable motions. As for the "rules," we opted to borrow them from the six traditional Korean games that were also played in the Netflix TV series "Squid Game" (Hwang, 2021). These games involve substantial amounts of physical movements, which naturally provide room for a diverse set of human poses and motions. The diversity is further influenced by in-game factors such as the uniquely defined rules of each game, the number of players, and the interactions between them.

Under our rule-guided motion design approach, each *scenario* run (i.e., one round of a game played with specific settings) in the virtual environment is initialized by carrying out a scenario design step which is followed by the incorporation of real-world motions. The scenario design involves the setup of all the knobs that control the appearance, players (winners and losers), game dynamics (e.g., how/when each game ends), and the human motion evolutions for each specific scenario. This is where the high-level rules of a given game are defined, and the coarse boundary of how human motions can evolve within the game is set. The incorporation of real-world motion is the phase where a rich variety of motions truly comes to life. Details on the entire pipeline will be elaborated in Section 3.

In addition, we also took into account that human appearance can vary greatly depending on the perspective from which it is viewed. Accordingly, we capture every scene from **multiple viewpoints** by implementing several image-capturing devices to take advantage of different perspective-related characteristics: three Unmanned Aerial Vehicles (UAVs), three Closed-Circuit Televisions (CCTVs), and one Unmanned Ground Vehicle (UGV). The three UAVs fly with random trajectories at different altitudes, the three CCTVs are located at the front, side, and back of the game playground, and the UGV moves randomly within the playground where the game is being played. These devices offer a variety of image-capturing properties, including aerial-*vs*-ground and dynamic-*vs*-static. Our strategy, designed to provide very diverse viewpoints in scene capture process, not only serves to ensure that the dataset includes more diverse human appearances but also broadens the potential tasks (*e.g.*, re-identification, multi-view applications, aerial-to-ground scene matching, *etc.*) for which the dataset can be used.

---

[1]This motion description was used to construct the Mocap Database HDM05 in AMASS (https://resources.mpi-inf.mpg.de/HDM05/05-01/index.html)

By leveraging the aforementioned human appearance-diversifying strategies, we construct a large-scale synthetic human dataset called *SynPlay* that contains more than *73k images with 6.5M human instances*, see sample images in Fig 1. To demonstrate SynPlay's ability to represent a variety of human appearances to the extent seen in the real world, we conduct a series of experiments where we evaluate the impact of leveraging SynPlay alongside real (non-synthetic) datasets curated for a variety of human-related tasks, *i.e.,*, aerial-view/ground-view human detection and segmentation. For all the tasks, training with SynPlay outperforms its counterparts (i.e., training-from-scratch, using other synthetic data) across a variety of datasets. Experiments also demonstrate that the SynPlay dataset significantly improves model performance on data-scarce tasks, highlighting its value in scenarios that require substantial supplementary training data.

## 2 RELATED WORKS

**Synthetic human data.** The creation of various synthetic human datasets has been facilitated by the advancement of modern synthetic data rendering engines such as Blender, Unity, and Unreal, alongside human modeling tools like MakeHuman (MakeHuman Community) and Character Creator (Studio Chacre). These rendering engines enable a realistic representation of humans in 3D virtual environments, while the modeling tools give creators precise control over the design of virtual characters. The creators of these datasets leveraged these tools to meticulously control key design factors, ensuring suitability for specific tasks, *e.g.*, SOMAset (Barbosa et al., 2018), PersonX (Sun & Zheng, 2019), UnrealPerson (Zhang et al., 2021), CARGO (Zhang et al., 2024) for Re-Identification, SURREAL (Varol et al., 2017) for pose estimation, GTA5 (Richter et al., 2016) for semantic segmentation, and Archangel-Synthetic (Shen et al., 2023b) for detection.

Recently, several attempts have been made to enhance the realism of virtual human models, with the aim of bringing them closer to the quality of their real-world counterparts. SMPL-X (Pavlakos et al., 2019) and AMASS (Mahmood et al., 2019) used motion capturing devices to capture natural human motions. BEDLAM (Black et al., 2023) tried to improve the diversity of various factors such as skin-tones or clothing that affect human outer appearance, still relying on the SMPL-X. However, motion scanners imposed constraints on the environment, particularly in capturing large motions or events involving multiple humans. While AGORA (Patel et al., 2021) and ScoreHMR (Stathopoulos et al., 2024) sought to digress from the usage of the motion scanners by fitting human body models to the human motions in real-world images/videos, the quality of the fitted human models declined drastically on the images taken from a distance. One of our goals for our dataset was to incorporate multiple viewpoints, including distant views of the target scene. To avoid compromising the quality of human motions/poses, we chose to use motion capture devices, while implementing our approach to ensure that final results in the dataset are not limited by the environments in which the devices were used.

**Natural human motion acquisition.** Whether we are creating a real or synthetic dataset, images of motions captured by directing humans to perform specific actions based on a description often appear awkward rather than natural. Because of that, most datasets aim to include humans engaged in daily activities (*e.g.*, MS COCO (Lin et al., 2014), MPII Human Pose (Andriluka et al., 2014)), performing tasks such as sports (*e.g.*, UCF-Sports (Rodriguez et al., 2008), SoccerNet (Cioppa et al., 2022), SportsMOT (Cui et al., 2023)) or art (*e.g.*, Human-Art (Ju et al., 2023)) to capture their motions and poses in the most natural states possible. However, it is self-contradictory to artificially create a virtual event to capture natural motions associated with the event. In this paper, we aim to detour this self-contradict by initially designing the virtual events (*i.e.*, aforementioned games) using existing but non-natural virtual motions, which are then replaced by real-world motions captured using a motion capture device.

**Supplemental datasets for training.** Enhancing model performance by supplementing the training with additional data has been a common strategy (Ren et al., 2016; Lee et al., 2019). Initially, this involved combining datasets constructed with the same purpose, like MS COCO (Lin et al., 2014) and PASCAL VOC (Everingham et al., 2015), for tasks such as object detection. Some approaches utilized large-scale datasets (*e.g.*, ImageNet (Deng et al., 2009) or Instagram (Mahajan et al., 2018)), which were not necessarily designed for the target task, to build foundational features, followed by a transfer learning such as pretrain-finetune (Girshick et al., 2016) or PTL (Shen et al., 2023a) to adjust the model on the target dataset. As models have grown in size and complexity (*e.g.*,

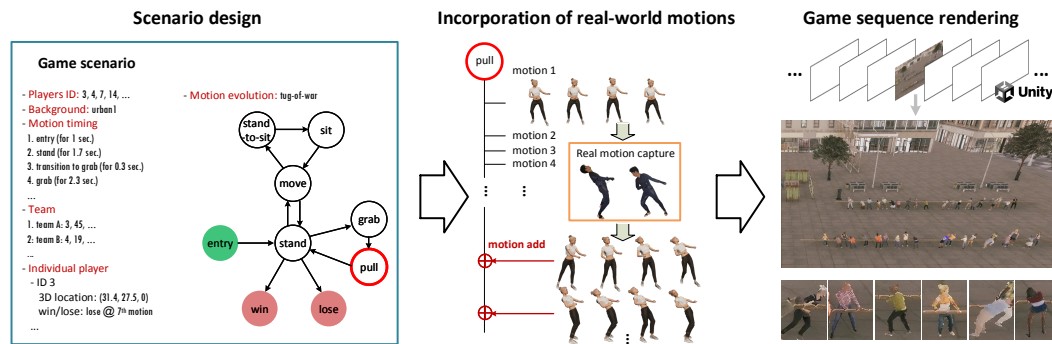

Figure 2: **Game sequence generation pipeline.** This illustrate how we create a sequence for a `tug-of-war` game which includes an example of how we incorporate real-world motions towards the elementary motion state of `pull`. In the motion evolution graph, the start and end nodes are indicated by green and red circles, respectively. A diverse set of `pull` motion instances is shown below the image of the rendered scene.

ViT (Dosovitskiy et al., 2021)), the demand for large-scale, high-quality datasets has increased, but the high costs of annotation present a significant barrier. To address this, label-agnostic training methods like self-supervised learning (Chen et al., 2020; Caron et al., 2020; Grill et al., 2020; He et al., 2022) and synthetic dataset generation with cost-free annotations have emerged as viable solutions. In response, we have specifically designed SynPlay to supplement various computer vision tasks that require a large-scale, highly-diversified human appearance set.

# 3 SYNPLAY DATASET

In a nutshell, we aim to create a synthetic dataset in a virtual environment that captures the diversity of human appearances found in the real world.

Accordingly, we design our dataset with a focus on two key aspects that naturally diversifies the human appearances captured in the scenes: i) expanding the motion set with increased reality, and ii) capturing each scene from a diverse set of camera viewpoints.

**Diverse yet realistic human motions.** We use *rule-guided motion design* in our SynPlay dataset, borrowing "rules" from six traditional Korean games, also featured in the Netflix series "Squid Game" (Hwang, 2021)[2]. This approach offers coarser motion guidance for the virtual players, facilitating the generation of a wide spectrum of natural motions, even including the ones that defy detailed description.

Our rule-guided motion design is effectively baked into the overall sequence-generating design pipeline, as shown in Figure 2, that consists of the *scenario design* followed by the *incorporation of real-world motions*. The scenario design involves the setup of all the knobs that control the appearance, players (winners and losers), game dynamics (*e.g.*, how/when each game ends), and the human motion evolutions for each specific scenario. All the items within each scenario that do not have to be hard-coded (*e.g.*, game rules) are selected randomly when designing each scenario. The motion evolution of each virtual player in a specific game is governed by a graph structure where all possible elementary motions and their potential transitions are represented as nodes and directed edges, respectively. Each node is tied with a pool of motions that fall under the same elementary motion state (*e.g.*, move, sit). As the game progresses, a virtual player evolves its motion following the directed edges and stays there according to the 'motion timing', also defined in the scenario design. At each state, the virtual human randomly chooses to exhibit one of the motions in the corresponding pool. Note that, while a uniquely designed scenario is used for a unique sequence, the same motion evolution graph is used for all the sequences captured under the same game rule.

---

[2]Our game scenarios are designed based on the traditional game rules, without taking any specific situations from the show.

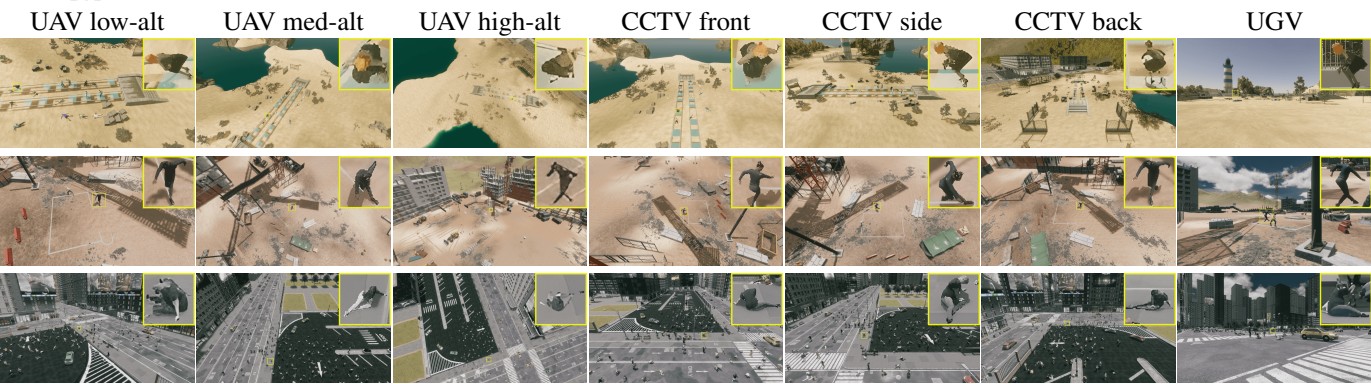

| UAV low-alt | UAV med-alt | UAV high-alt | CCTV front | CCTV side | CCTV back | UGV |
|---|---|---|---|---|---|---|

Figure 3: **Multiple viewpoints used in SynPlay.** On the top-right corner of each image, we place the enlarged crop of one human instance who is visible from all seven viewpoints in each scenario. Multiple camera viewpoints allow substantial variations in appearance for the same human subject with identical pose.

Before incorporating the real-world motions, we leverage two techniques to pre-diversify the elementary human motions readily available in human motion libraries such as Mixamo Adobe: i) dynamically blending two existing motions of similar types to generate a new motion type (*e.g.*, blending slow-walking and running to generate hasty-walking), ii) using elementary motions as animation layers to make a new motion (*e.g.*, raising hands while walking). On top of the pre-diversified set of motions for each game, a real human player wearing a motion capture device [3] is asked to either similarly mimic or newly create motions that align with the given game rule (thus, *rule-guided*). For example, for the game of tug-of-war, human players were provided with the game *rule*, and then asked to reenact any possible motion with the freedom of choosing the winning or losing side. For some games, more than one players were asked to actually play the game together to capture the motions that can naturally arise at the time of physical interactions. As the result of *incorporating the real-world motions*, the total number of unique motions in SynPlay increased from 104 to 257.

**Multiple viewpoints.** The camera viewpoints within SynPlay are diversified by implementing three widely used types of image-capturing platforms in the real world: UAV, UGV, and CCTV. They cover a variety of image-capturing properties like static/dynamic and ground/aerial. Viewpoint diversity is acquired by controlling the locations and the focal points of the cameras. Three UAVs, three CCTVs, and one UGV have been deployed (Figure 3), resulting in seven unique viewpoints for every game sequence. The UAVs are deployed to fly at various random locations while maintaining altitudes of low ($\sim30m$), medium ($\sim50m$), and high ($\sim100m$). CCTVs are located at a height of $15m$ at the front, back, and either side of the game playground. UGV images are captured assuming that a vehicle is randomly roaming on the ground. The focal points are set at several locations close to the area where the game usually takes place. For the UAVs, the focal point is changed to a random location every 10 $sec$, where each change takes 5 $sec$ to be fixed at a location for another 5 $sec$. Focal points of the CCTVs and the UGV do not change once determined.

**Dataset specification.** We created ten scenarios for each game, resulting in 60 game scenarios in total. Frames were rendered from seven different camera viewpoints at 1 fps with the resolution of 1920×1080, resulting in a total of 73,892 images with more than 6.5M human instances. The frame generation rate was selected as 1 fps to avoid including highly redundant human poses. Taking full advantage of the game engine's ability to generate annotations while rendering the scenes, we provided various types of ground truth annotations useful for various computer vision tasks: 2D/3D bounding boxes, instance-level segmentation masks, depth maps, and human keypoint locations.

## 3.1 OTHER DESIGN FACTORS

**Characters.** We have designed 456 virtual characters using the Character Creator, where each character is involved in multiple game scenarios. To vary the appearance of the characters and avoid

---

[3]Each real player used a SmartSuit Pro II and a pair of SmartGloves from Rokoko (http://rokoko.com)

generating biases, each character was uniquely designed with gender, skin color, age, height, obesity (body type), hair (styles and colors), and outfits. We kept the gender ratio between male and female at 1:1 and the ratio of skin color among white, black, yellow, and brown at 1:1:1:1. For age, each character was designed to fall into one of three categories: child, middle-aged, and elderly, and the ratio was set at 1:2:1. For each gender and age group, heights were modeled to follow a bell-shaped distribution, resulting in an overall dataset range of 140 to 190 cm. We manually designated every character with a unique outfit, while setting the hair and obesity aspect as diverse as possible.

**Backgrounds.** For each scenario, we set different environmental factors: sites, lighting conditions, and weather. A game takes place at one of ten custom-built sites. These include five urban locations (three common city areas, a construction site, and a factory site) and five natural sites (a green area, a snowy field, a desert, a meadow, and a beach). Multiple locations within each site map can be used as local playgrounds. To vary the lighting conditions, we take into account five different times of day: dawn, morning, noon, afternoon, and sunset. We consider three types of weather: sunny, foggy, and rainy. All of these are randomly determined for each scenario.

## 4 TASK EVALUATION

In line with the inherent purpose of synthetic data to serve as supplemental training data, we use the entire SynPlay dataset to train models for various computer vision tasks and evaluate its positive impact towards task performance. Our major counterpart models in evaluation are *trained-from-scratch*, which are trained only on real images (denoted as real in evaluation tables). We also validate the advantage of using SynPlay over other synthetic datasets.

### 4.1 GENERAL TASKS: DETECTION AND SEGMENTATION

We evaluate the SynPlay dataset on two general vision tasks, human detection and segmentation. These tasks require the ability to identify diverse human appearances in images captured at a distance. To leverage synthetic data during training, we adopt a *pretrain-finetune* strategy, where a model is pre-trained on synthetic data and fine-tuned on target real-world data. The detectors used in the experiments are YOLO v8 models (Jocher et al., 2023) with three different architecture sizes (small, medium, and large). Mask2former (Cheng et al., 2022) with the Swin-Base (Liu et al., 2021) backbone was used for segmentation. For evaluation metrics, we use the COCO-style APs which are two bounding box APs ($AP^{bb}$ and $AP^{bb}_{50}$)[4] for human detection and Intersection-over-Union (IoU) for the segmentation.

The main tasks are conducted on aerial-view datasets, which feature a wider range of human appearances, making them ideal for validating the design philosophy behind the SynPlay dataset. We also conduct experiments on ground-view datasets to evaluate SynPlay on a more widely studied task in the community.

**Aerial-view tasks.** Table 1 shows the results for aerial-view human detection and semantic segmentation tasks. Overall, for both tasks, using SynPlay in training provides remarkably better accuracy than all the compared cases, including real and all the other variations involving other synthetic data.

Notably, the results show that warming up the model with synthetic data before incorporating real data generally does not improve performance, except in the case of SynPlay. In other words, unless the synthetic dataset is properly designed and constructed, we cannot expect performance improvement simply from adding synthetic data to the training process. Moreover, among cases using synthetic data only in training, SynPlay presents unparalleled accuracy. In fact, the results using other sources of synthetic data are so poor that the other sources can be considered useless for this type of dataset utilization. Based on these two observations, *our design strategies for enhancing the diversity and realism of human appearance are shown to be highly effective in meeting expectations*.

**Ground-view tasks.** Table 2 explores the impact of using SynPlay for the general computer vision tasks of ground-view human detection and semantic segmentation. We also evaluate how models perform when only the subset with matching viewpoint (i.e., UGV images in SynPlay) is used in training. Overall, using the entire SynPlay yields the highest accuracy on both tasks, while using

---

[4]Detection accuracies in the following tables are reported with two numbers in the form of $AP^{bb}/AP^{bb}_{50}$.

Table 1: **Comparison with other synthetic datasets on aerial-view human detection and semantic segmentation.** The numbers in parentheses are the gaps from the model trained without synthetic data ('real'). Positive and negative gaps are indicated in green and red fonts, respectively. The best accuracy for each setting is shown in **bold**. *Notations*: '+ real' represents a model pre-trained with synthetic data and fine-tuned on a 'real' dataset, where 'real' is a training set of dataset used for evaluation. 's', 'm', and 'l' represent three YOLO v8 models with different architectures.

| | (a) human detection | | | | | | | | | (b) semantic seg. | |
|---|---|---|---|---|---|---|---|---|---|---|---|
| data in training | VisDrone (Zhu et al., 2022) | | | Okutama-action (Barekatain et al., 2017) | | | Semantic Drone (ICG) | | | Semantic Drone (ICG) | Aeroscapes (Nigam et al., 2018) |
| | s | m | l | s | m | l | s | m | l | | |
| real | 19.72/47.43 | 21.14/49.52 | 21.60/51.10 | 27.40/75.17 | 28.99/76.60 | 31.53/78.78 | 44.00/ 77.20 | 44.52/ 78.52 | 42.62/ 79.87 | 0.66 | 22.25 |
| Archangel (Shen et al., 2023b) | 0.23/ 0.63 | 0.38/ 0.98 | 0.59/ 1.48 | 2.59/ 8.45 | 3.90/10.13 | 2.83/ 9.12 | 0.64/ 1.59 | 2.42/ 5.37 | 0.94/ 1.62 | 0.74 | 0.04 |
| SynDrone (Rizzoli et al., 2023) | 0.31/ 0.81 | 0.36/ 0.84 | 0.71/ 1.89 | 0.00/ 0.00 | 0.00/ 0.01 | 0.00/ 0.00 | 0.00/ 0.00 | 0.00/ 0.00 | 0.00/ 0.00 | 0.07 | 0.00 |
| **SynPlay** | 5.29/11.75 | 4.31/ 9.12 | 2.79/ 5.87 | 12.74/40.86 | 8.19/25.43 | 8.15/25.23 | 7.02/ 12.21 | 9.60/ 15.51 | 15.71/ 23.59 | 8.03 | 6.44 |
| Archangel + real | 18.77/45.39 | 20.25/48.52 | 20.82/49.51 | 30.72/80.35 | **32.36**/80.63 | 31.71/79.63 | 46.60/ 74.07 | 48.60/ 75.86 | 44.62/ 73.23 | 9.28 | 20.61 |
| | (-0.95/-2.04) | (-0.89/-1.00) | (-0.78/-1.59) | (+3.32/+5.18) | (+3.37/+4.03) | (+0.18/+0.85) | ( +2.60/ -2.13) | ( +4.08/ -2.66) | ( +1.00/ -6.64) | ( +8.62) | (-1.64) |
| SynDrone + real | 18.78/45.79 | 20.94/49.44 | 21.97/51.51 | 29.70/77.71 | 31.39/79.42 | 31.24/78.71 | 50.93/ 82.28 | 53.71/ 85.47 | 59.59/ 85.02 | 5.56 | 24.59 |
| | (-0.94/-1.64) | (-0.20/-0.08) | (+0.37/+1.41) | (+2.30/+2.54) | (+2.40/+2.82) | (-0.29/-0.07) | ( +6.93/ +5.08) | ( +9.19/ +6.95) | (+16.97/ +5.15) | ( +4.90) | (+2.34) |
| **SynPlay + real** | **20.88/49.31** | **22.34/52.12** | **22.98/52.93** | **32.47/81.60** | 31.96/**81.13** | **33.17/82.52** | **66.52/ 90.33** | **69.46/ 91.35** | **68.82/ 91.37** | **23.32** | **32.19** |
| | (+1.16/+1.88) | (+1.20/+2.60) | (+1.38/+1.83) | (+5.07/+6.43) | (+2.97/+4.53) | (+1.64/+3.74) | (+22.52/+13.13) | (+24.94/+12.83) | (+26.20/+11.50) | (+22.66) | (+9.94) |

Table 2: **Impact of SynPlay on MS COCO** (`person` category). *Notation*: 'SynPlay-UGV' and 'SynPlay-all' are a UGV subset of SynPlay and the entire SynPlay, respectively.

| | (a) human detection | | | (b) sem.seg. |
|---|---|---|---|---|
| data in training | s | m | l | |
| real | 46.19/65.91 | 50.10/69.86 | 52.52/72.15 | 15.10 |
| **SynPlay-UGV + real** | 46.53/66.18 | 50.70/70.37 | 52.69/72.29 | 20.18 |
| | (+0.34/+0.27) | (+0.60/+0.51) | (+0.17/+0.14) | (+5.08) |
| **SynPlay-all + real** | **46.84/66.70** | **51.12/70.74** | **53.00/72.59** | **21.57** |
| | (+0.65/+0.79) | (+1.02/+0.88) | (+0.48/+0.44) | (+6.47) |

Table 3: **Synergy impact with MS COCO** on aerial-view human detection. YOLO v8 model with a medium size architecture is used.

| data in training | VisDrone | Okutama-action | Semantic Drone |
|---|---|---|---|
| real | 21.14/49.52 | 28.99/76.60 | 44.52/78.52 |
| COCO | 7.16/16.46 | 15.17/48.28 | 34.74/56.39 |
| SynPlay | 4.31/ 9.12 | 8.19/25.43 | 9.61/15.52 |
| COCO + SynPlay | 11.49/25.20 | 14.68/49.82 | 18.60/31.03 |
| COCO + real | 22.11/51.73 | 32.26/80.10 | 65.72/89.20 |
| SynPlay + real | 22.34/52.13 | 31.96/**81.13** | 69.46/91.35 |
| COCO + SynPlay + real | **22.78/53.01** | **33.82**/79.44 | **73.52/92.80** |

the UGV-subset still outperforms the model trained without SynPlay. These results demonstrate that our insight in ensuring diversity by varying the camera viewpoints is effective even in tasks that do not contain such multiple viewpoints. In addition, the greater improvement in semantic segmentation over object detection shows that ensuring diversity is more effective in tasks that require more detailed human representation models.

**Combination with MS-COCO for pre-training dataset.** The effect of using pre-training can be greater when applying two or more datasets with complementary properties. Here, we aim to investigate the potential synergy achieved by integrating MS COCO, a real dataset primarily comprising ground-view images, with SynPlay for the task of aerial-view human detection. Table 3, shows all combinations of SynPlay and MS COCO datasets when used for pre-training. The anticipated synergistic effect appears in all cases except in one case ($AP_{50}^{bb}$ results on Okutama-action) when fine-tuned on the target dataset. Moreover, using *SynPlay only* provides comparable accuracy to using MS COCO when used indirectly through fine-tuning on the real dataset.

Interestingly, the results without fine-tuning show a different trend. On Okutama-action and Semantic Drone cases, using MS COCO performed better than other two baselines, while SynPlay specifically showing a much lower accuracy. We observe that synthetic data still lags behind real-world data in many respects, highlighting the need for further research to bridge the gap.

## 4.2 DATA-SCARCE TASKS: FEW-SHOT AND CROSS-DOMAIN LEARNING

In this section, we compare SynPlay with other synthetic datasets on its ability to meet the demand for additional data in data-scarce tasks. For data-scarce tasks, we adopt few-shot and cross-domain learning tasks on aerial-view human detection, which suffers more severely from a lack of training data than ground-view detection. Following the data-scarce task setups of Shen et al. (2023a), we train models with two few-shot regimes using 20 and 50 images of VisDrone (denoted by 'Vis-20/50'). We test the models on VisDrone, Okutama-action, and Semantic Drone where the evaluations done on the last two datasets can be seen as 'cross-domain'. To attenuate the potential random

Table 4: **Few-shot and cross-domain learning accuracy** on aerial-view human detection. To clarify, the accuracy on Okutama-action and Semantic Drone refers to cross-domain learning performance. *Notation*: 'Archangel*' is an expanded 'Archangel' to be pose-diversified (Shen et al., 2024).

| data in training | method | Vis-20 | | | Vis-50 | | |
|---|---|---|---|---|---|---|---|
| | | VisDrone | Okutama-action | Semantic Drone | VisDrone | Okutama-action | Semantic Drone |
| real | | 0.58/ 2.27 | 3.64/ 14.54 | 0.62/ 1.89 | 0.76/ 3.30 | 7.82/ 28.66 | 1.30/ 5.65 |
| + Archangel | PTL | 2.07/ 6.72 | 7.90/ 31.53 | **8.81/ 33.71** | 2.92/ 9.26 | 11.49/ 42.51 | **8.98/ 33.21** |
| + Archangel* | | 2.26/ 7.39 | 8.95/ 36.97 | 6.45/ 26.13 | 2.99/ 9.42 | 12.89/ 47.24 | 6.29/ 25.50 |
| **+ SynPlay** | | **3.08/ 9.03** | **14.39/ 49.53** | 6.94/ 24.22 | **3.71/11.20** | **15.67/ 52.06** | 7.74/ 26.99 |
| | | (+2.50/+6.76) | (+10.75/+34.99) | ( +6.32/+22.33) | (+2.95/+7.90) | (+7.85/+23.40) | (+6.44/+21.34) |
| + Archangel | PT-FT | 0.76/ 2.48 | 4.24/ 17.17 | 6.53/ 23.67 | 1.29/ 3.76 | 5.32/ 20.96 | 7.10/ 27.95 |
| + Archangel* | | 1.21/ 4.02 | 9.14/ 34.70 | 8.20/ 28.80 | 1.84/ 5.37 | 10.39/ 36.83 | 8.63/ 30.09 |
| **+ SynPlay** | | **2.94/ 9.38** | **12.19/ 40.88** | **11.32/ 37.30** | **3.72/11.87** | **13.66/ 44.06** | **12.76/ 41.66** |
| | | (+2.36/+7.11) | ( +8.55/+26.34) | (+10.70/+35.41) | (+2.96/+8.57) | (+5.84/+15.40) | (+11.46/+36.01) |

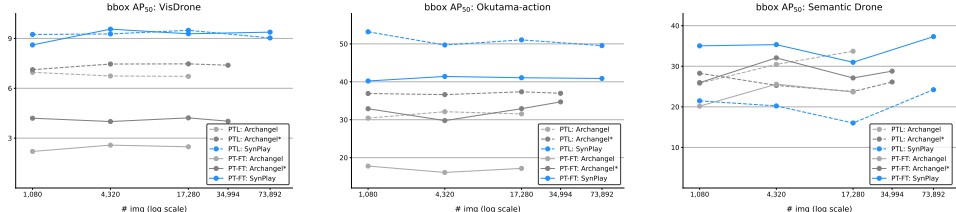

Figure 4: **Scaling behavior of synthetic datasets** under the Vis-20 setup ($AP_{50}^{bb}$). Scaling behavior of each dataset is compared by randomly sampled subsets of 1,080, 4,320, and 17,280 images, which correspond to 1/16th the size, 1/4th the size, and the size of Archangel. For reference, the sizes of Archange* and SynPlay are 34,994 and 73,892, respectively.

effects that may arise when selecting real training images, all reported numbers are average accuracy over three runs.

As baseline methods leveraging synthetic data in training, we use a pretrain-finetune strategy (PT-FT) and Progressive Transformation Learning (PTL) (Shen et al., 2023a). PTL is a progressive data augmentation approach that iteratively expands the training set by adding a subset of synthetic data, which is transformed to look real. In each PTL iteration, a subset of the synthetic data is selected, such that synthetic data that is closer to the real dataset is selected more often. For the data-scarce tasks experimented in Shen et al. (2023a), PTL was better than PT-FT while both outperformed the cases without synthetic data. We used RetinaNet (Lin et al., 2017) as the detector.[5]

**Comparison with other synthetic data.** In Table 4, we compare the detection accuracy of the models trained with different synthetic datasets on the few-shot and cross-domain learning tasks. With PT-FT, SynPlay achieved significantly better accuracy than other synthetic datasets across all three datasets. With PTL, SynPlay performed the best on VisDrone and Okutama-action for both Vis-20 and Vis-50 settings. Even on Semantic Drone, which shows an unusual performance trend, the best performance was achieved when SynPlay was used via PT-FT.

In addition, compared to SynPlay's performance improvement on general tasks (in Table 1), the improvement achieved on data-scarce setting by SynPlay is much greater on VisDrone and Okutama-action on data-scarce tasks. This demonstrates that *SynPlay effectively meets the demand for additional data in data-scarce setting*, which is greater than that in general tasks. We will discuss the unexpected performance trends on Semantic Drone in more detail in Sec. 5.

**Scaling behaviors.** To fairly validate the performance comparison without being affected by dataset size, we explore the scaling behavior of synthetic datasets. In Fig 4, we compare the detection accuracy of three synthetic datasets at multiple points where the datasets are randomly sampled to have the same size. On all three test sets, the best performing models use SynPlay in training, *i.e.*, SynPlay + PTL on VisDrone and Okutama-action, SynPlay + PT-FT on Semantic Drone. *The performance gain achieved using SynPlay is not simply due to the large size of the dataset.*

---

[5]PTL was designed to be suitable for RetinaNet. For a fair comparison between PTL and PT-FT, we used RetinaNet instead of YOLO v8 for this experiment.

Table 5: **FID comparison.** In FID calculation, VisDrone serves as a reference representing real aerial-view human data.

| COCO | Archangel | Archangel* | SynDrone | **SynPlay** |
|---|---|---|---|---|
| 48.16 | 67.20 | 67.20 | 21.66 | **18.36** |

Table 6: **Proportion of nadir-view instances in synthetic datasets** used in data-scarce tasks. Instances with camera viewing angle from ground greater than 71.57° are considered as nadir-view instances.[6]

| Archangel | Archangel* | SynPlay |
|---|---|---|
| 25.00% | 12.82% | 4.24% |

## 4.3 IMAGE QUALITY EVALUATION

In Table 5, for all training datasets involved in our experiments, we calculate FID (Fréchet Inception Distance) (Heusel et al., 2017) to assess their fidelity and diversity. SynPlay presents the best score compared to other synthetic datasets, which is a result that aligns well with our task results. These results suggest that SynPlay's superior task performance is achieved by better fidelity and diversity, which are our goals in designing SynPlay. Moreover, the better FID of SynPlay compared to MS COCO, which mainly includes ground-view images, is also reported, supporting the hypothesis that adopting multiple viewpoints effectively diversifies human appearances.

## 5 DISCUSSION AND CONCLUSION

**Peculiar performance trend on Semantic Drone.** Following conflicting phenomena were observed in experimental results when tested on Semantic Drone:

- *When using synthetic data in training via PTL for data-scarce tasks, involving SynPlay under-performed when compared to cases using other synthetic datasets.*

- *The performance gain acquired by incorporating a synthetic data during training is remarkably large for Semantic Drone when compared to the other two cases, while SynPlay's incorporation showcasing the most substantial gain.*

Most human instances in Semantic Drone are taken from nadir views, while VisDrone and Okutama-action have instances captured with fewer nadir-views. The portion of nadir-view instances in SynPlay is the smallest among the synthetic datasets used in data-scarce tasks (Table 6). As PTL continues to prioritize samples from synthetic data that closely resemble the seed data (*i.e.,* VisDrone) for training, the reduced selection of nadir-view instances from the SynPlay may result in a lower gain (first phenomenon). On the other hand, the second phenomenon indicates that ensuring greater diversity via using supplemental synthetic data has greater impact on Semantic Drone, which lacks diversity due to its limited viewpoints. Moreover, SynPlay that is less similar to Semantic Drone while being more diverse in relation to the compared synthetic datasets, shows the largest impact, supporting our claim that improving the diversity is generally effective in constructing a better synthetic data.

**Conclusion.** *What motion a human performs* and *where a person is viewed from* are two crucial factors that make a difference in *how a human looks*. We create a synthetic human dataset called *SynPlay* with the aim of expanding the diversity human appearance by diversifying these factor. Ensuring the diversity allowed SynPlay to have a greater positive impact towards model training when compared to other counterparts (train-from-scratch, using other synthetic data) on aerial-view/ground-view object detection and semantic segmentation. This positive impact of SynPlay becomes even greater in data-scarce tasks, where synthetic data is strongly desired as supplemental training data.

---

[6]This criterion is the largest viewing angle from ground on Archangel*.

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

## APPENDIX

## A    COMPARISON TO OTHER DATASETS

Table 7 Table 1 provides a comparative analysis of various human datasets, categorized as real or synthetic and captured from either ground or aerial perspectives. Key observations from this comparison are outlined below:

1. Aerial-view sets, *thanks to their wide viewing angles*, generally have *more human instances per image* than ground-view sets, except for few cases that employed a fixed number of actors in a real set or designing one instance per image in a synthetic set.

2. Aerial-view sets generally contain a wider range of viewpoints. (mostly near∼far)

3. For existing synthetic datasets, aerial-view sets typically feature *fewer motion variations* compared to ground-view sets. This is because *aerial-view datasets often prioritize leveraging a wide range of viewpoints over expanding the variety of human motions.*

Table 7: **Comparison of human datasets.** '#inst/img' is acquired only on images that contain human. '#motion' indicates the number of unique motions depicted in the dataset, except the ones with the subscript 'pose' which indicate the number of static poses. Since a single motion can consist of multiple number of unique poses, #motion is generally smaller than the number of poses. For certain datasets, the test set without available labels is excluded from this comparison. 'uncountable' indicates that the number of human motions included in the set is countless/uncountable.

| dataset | domain | #inst | #img | #inst/img | natural motion | #motion | viewpoint |
|---|---|---|---|---|---|---|---|
| *ground-view* | | | | | | | |
| VOC 12 (Everingham et al., 2015) | real | 10K | 11.5K | 2.48 | daily | uncountable | near |
| KITTI (Geiger et al., 2012) | real | 4.5K | 7.5K | 2.52 | daily | 2 | near |
| COCO Dev17 (Lin et al., 2014) | real | 649K | 164K | 9.72 | daily | uncountable | near |
| MPII Human Pose (Andriluka et al., 2014) | real | 40K | 24.9K | 1.61 | daily | 20 | near |
| Cityscapes (Cordts et al., 2016) | real | 21.4K | 5K | 7.85 | daily | 2 | near |
| ADE20K (Zhou et al., 2017) | real | 30K | 27.5K | 4.36 | daily | uncountable | near |
| Human-Art (Ju et al., 2023) | real | 123K | 50K | 2.46 | art | uncountable | near |
| GTA5 (Richter et al., 2016) | synth | 1.4M | 1.4M | 1 | ✗ | $20K_{pose}$ | near |
| SURREAL (Varol et al., 2017) | synth | 6.5M | 6.5M | 1 | detail+mocap | 23 | near |
| SOMAset (Barbosa et al., 2018) | synth | 100K | 100K | 1 | detail+mocap | $250_{pose}$ | near |
| PersonX (Sun & Zheng, 2019) | synth | 273K | 273K | 1 | ✗ | $4_{pose}$ | near |
| UnrealPerson (Zhang et al., 2021) | synth | 120K | 120K | 1 | ✗ | 2 | near |
| AGORA (Patel et al., 2021) | synth | · | 19K | 1∼15 | detail+mocap | $4,240_{pose}$ | near |
| BEDLAM (Black et al., 2023) | synth | · | 380K | 1∼10 | detail+mocap | $2,311_{pose}$ | near |
| *aerial-view* | | | | | | | |
| Okutama-action (Barekatain et al., 2017) | real | · | 77K | ∼9 | detail | 12 | med |
| Semantic Drone (ICG) | real | 1.5K | 400 | 4.16 | daily | unspecified | med |
| UAVid (Lyu et al., 2020) | real | 4.7K | 420 | 20.06 | daily | unspecified | med∼far |
| VisDrone (Zhu et al., 2022) | real | 109K | 40.0K | 15.42 | daily | unspecified | med |
| Archangel-real (Shen et al., 2023b) | real | 165.6K | 41.4K | 4 | detail | $3_{pose}$ | near∼far |
| Archangel-mannequin (Shen et al., 2023b) | real | · | 178.8K | 6∼7 | detail | $3_{pose}$ | near∼far |
| Archangel-synth (Shen et al., 2023b) | synth | 4.4M | 4.4M | 1 | ✗ | $3_{pose}$ | near∼far |
| SynDrone (Rizzoli et al., 2023) | synth | 803K | 72K | 11.15 | ✗ | 2 | med∼far |
| CARGO (Zhang et al., 2024) | synth | 108K | 108K | 1 | ✗ | 2 | near∼far |
| **SynPlay** | synth | 6.5M | 73K | 88.40 | rule+mocap | uncountable | near∼far |

\* natural motion
· daily: human motions engaged in daily activity
· art: human motions shown in works of art
· detail: human motions captured by 'detail-guided design'
· rule: human motions captured by 'rule-guided design'
· +mocap: human motions captured using a motion scanner

4. Rule-guided design can utilize *significantly larger range of human motions* compared to detail-guided design.

The comparison shown in the table also demonstrates that SynPlay successfully addresses the shortfall of aerial-view synthetic datasets (3rd observation), while maximizing the benefits of aerial-view datasets (1st and 2nd observations).

Moreover, the 4th observation supports that our proposed rule-guided design is successful in securing the diversity of human motions in the set. It is noteworthy that while SURREAL (Varol et al., 2017) (constructed with 'detail+mocap') contains a comparable number (6.5M) of human instances as SynPlay, the number of motions manifested in the dataset is extremely limited when compared to SynPlay (23 vs. uncountable).

# B SynPlay Statistics

Here, we provide several statistics from the SynPlay. Fig 6 shows the distribution of bounding box sizes over human instances captured by each device. The majority of bounding box sizes are small, which illustrates a characteristic of aerial-view datasets. Interestingly, UAVs can capture human instances with larger bounding boxes than CCTVs. This could be due to the fact that, although UAVs are typically positioned at higher altitudes than CCTVs, there are more cases where UAVs get closer to real-time events and human instances, different from the fixed CCTVs.

Figure 5: **456 virtual players in SynPlay** created using Character Creator.

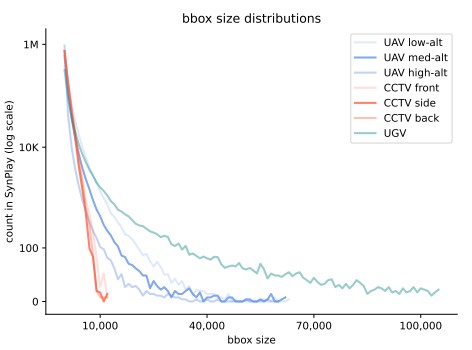

Figure 6: **Bounding box size distribution** for each image-capturing device.

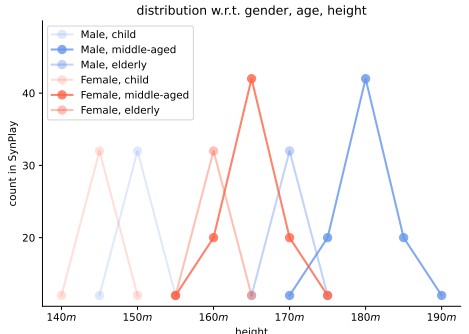

Figure 7: **Character height distribution** that varies according to gender and age.

Fig 7 shows the distribution of human height with respect to gender and age. As mentioned in the main manuscript, each distribution is formed as being bell-shaped. We create 456 virtual characters by controlling human height, gender, and age according to these distributions and uniquely diversifying other factors (skin color, obesity, hair, outfit, *etc*) as much as possible, as shown in Fig 5.

## C    IMPLEMENTATION DETAILS

### C.1    MOTION EVOLUTION GRAPH

Fig 8 shows motion evolution graphs used in designing the game scenarios for the SynPlay dataset. Even within the same game, the scenario may change, but the motion evolution graph will remain consistent. It is noteworthy to mention that, despite the wide range of situations and a variety of motions involved in the games, the motion evolution graph for each game consists of only a few motion nodes and their transitions. Given that each node (represented as an *elementary motion state* in the main manuscript) encompasses a range of motions, this illustrates the essence of a rule-based design approach where only basic game rules are provided to freely allow the diverse array of human motions to be manifested.

### C.2    EXPERIMENTAL SETTING

In our experiments, our goal is to explore the capabilities of SynPlay as supplementary training data on a variety of tasks. We mostly adhere to the original settings and implementations of the methods used in our experiments, with minimal modifications. The specific modifications used in our experiments are described below.

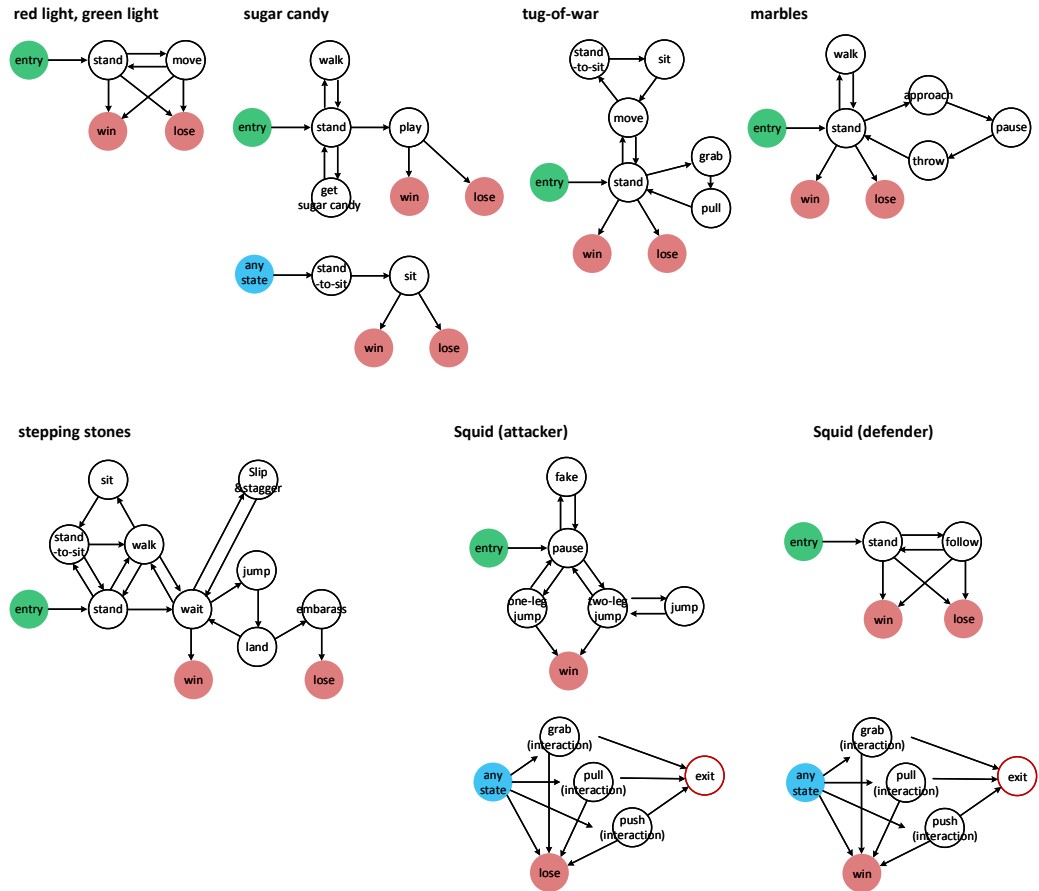

Figure 8: **Motion evolution graphs.** The start node ('entry') and the end nodes ('win' or 'lose') are indicated by green and red circles, respectively. For the games where secondary graphs are available (i.e., sugar candy or squid), at any given time (except at the start or end node), the current state in the main graph can move to the 'any state' node (blue-filled circle) in the secondary graph. When the 'end' node (red-bordered circle) is reached within the secondary graph, the current state moves its way back to the latest node that was touched in the main graph before entering the secondary graph.

**Architecture modification.** Our tasks, specifically human detection and semantic segmentation, can be viewed as one-class problems. Therefore, all method architectures, particularly the dimensions of the last layer, have been adjusted accordingly.

**Image size applied in YOLOv8 training/inference.** We use the image size of 1280×1280 for all datasets except for COCO, which uses an image size of 640×640. This decision simply takes into account the original image size of the datasets. Even after rescaling, the size range of human instances in the compared datasets remains similar. When using other models, *i.e.,* Mask2Former in semantic segmentation tasks and retinaNet in data-scarce tasks, the image size/scaling recommended in the original settings was used.

**Training Mask2Former without the large-scale jittering (LSJ) augmentation (Ghiasi et al., 2021).** We did not use the default LSJ augmentation in training the Mask2Former segmentation models solely for performance reasons. In all cases, segmentation accuracy were found to be significantly lower when LSJ augmentation was used. LSJ augmentation, which greatly expands the range of image scaling, may not be suitable for aerial-view detection, which mainly includes small-sized human instances. This performance degradation with LSJ augmentation is also observed in He et al. (2022), which is a reputable literature in the field of self-supervised learning.

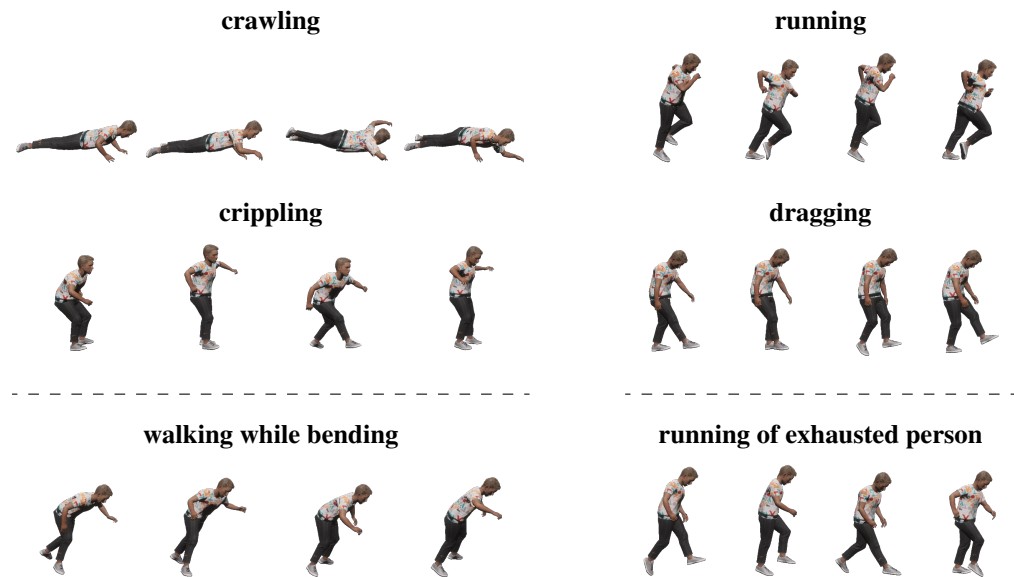

Figure 9: **Two motion blending examples**. For each example (left or right column), the two motions (top and middle row) is blended together to generate a new motion (bottom row). The blending ratio between the two input motions can be controlled. The names for the motions are not computationally involved in the blending process.

**Settings for PT-FT.** When using PT-FT in the general tasks, training specifications, including training epochs and learning rate, did not differ between pre-training and fine-tuning. In data-scarce tasks, we follow all the settings of Shen et al. (2023a) as outlined in PTL, while leaving out the progressive component.

**Settings for data-scarce tasks.** For all experiments performed for data-scarce tasks including the scaling behavior study, we follow all the settings and experimental environments of Shen et al. (2024).

### C.3 QUANTITATIVE MEASURES

We provide the detail on how we calculate the two metrics used for the quantitative analysis in the main manuscript.

**Fréchet Inception Distance (FID) (Heusel et al., 2017).** We utilized the PyTorch implementation of FID in Seitzer (2020) with the default setup to assess the fidelity and diversity for all the training datasets involved in our experiments. We did not perform image scaling on the input for any dataset, and the final average pooling features were used to compute FID.

**Proportion of nadir-view instances.** An instance with an elevation angle greater than $71.57°$ relative to the UAV as is considered to be a nadir-view instance, representing the maximum elevation angle for Archangel* (Shen et al., 2024). To identify nadir-view instances for Archangel, we utilized the dataset metadata, i.e., UAV position. Similarly, for Archangel*, we determined if an instance was an nadir-view instance based on the source instance, also using the dataset metadata. In the case of SynPlay, we computed the elevation angle for each instance using the absolute 3D coordinates of the instance and the UAV provided by SynPlay.

**sitting**     **walking**     **kneeling down**

**raising something up
and looking at it**     **throwing something**     **cheering**

**sitting, raising something up
and looking at it**     **throwing something
while walking**     **cheering
while kneeling down**

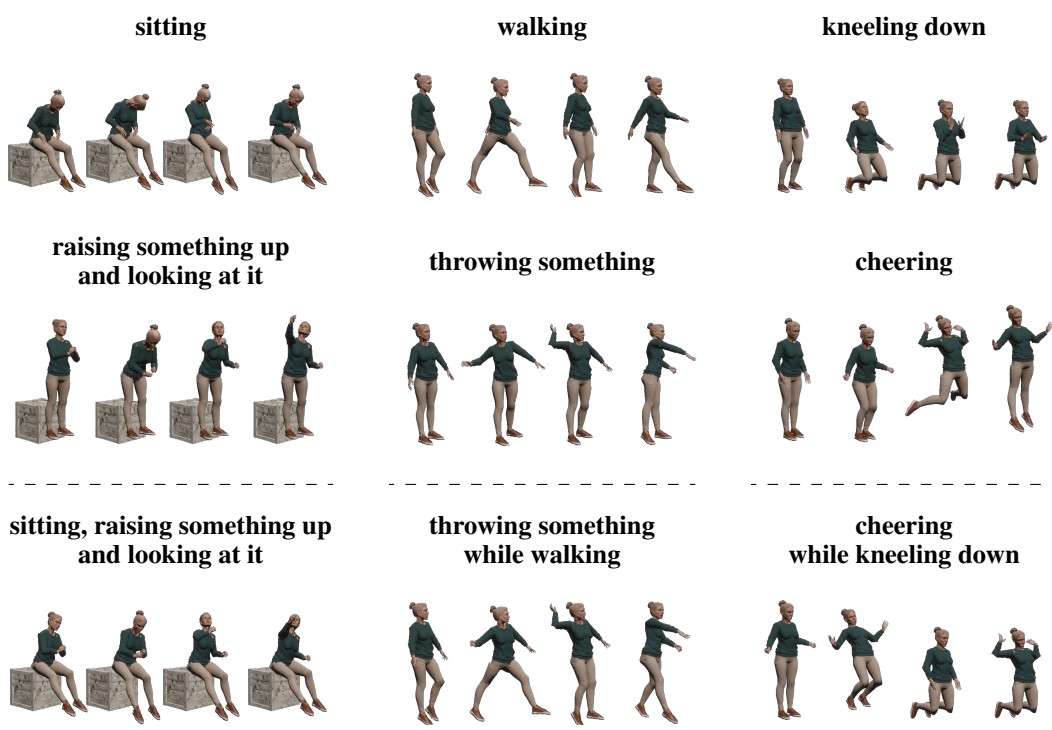

Figure 10: **Three examples of leveraging animation layers**. For each example (left, middle, or right), the resulting motion of leveraging the animation layers over two input motions (top and middle rows) is shown in the bottom row. Note that the semantic labels (e.g., walking, cheering) were not provided at the time of capture; they are included in the figure only for the convenience of the readers.

# D   QUALITATIVE ANALYSIS

## D.1   BLENDING AND ANIMATION LAYER

Fig 9 and 10 show several examples of the blending process and how the animation layers are leveraged: the two techniques for expanding human motions within the virtual environments, respectively. Interestingly, the motions created by blending is largely different from their corresponding input motions, while the motions created via leveraging the animation layers still exhibit the appearances and dynamics resembling both the input motions. These two techniques are readily available for use within the Unity environment.

## D.2   VIRTUAL MOTION AND REAL-WORLD MOTION

Fig 11 shows several examples of real-world motions. Real-world motions are created either by having the real human wearing the motion capture device mimic the pre-provided reference motions or by demonstrating potential in-game motions under the given game rules. It is observed that real-world motions can express a wider range of specific actions while maintaining a sense of realism. Moreover, motions that are difficult to pinpoint or describe can also be created, *e.g.,* multi-person wrestling motions.

## D.3   SYNPLAY SAMPLE IMAGES

Fig 12 includes additional sample images from the SynPlay dataset. Various human appearances depending on human motion differently taken according to the game scenario, and camera viewpoints are observed. Various human appearances are observed that change depending on human motions

taken differently according to the game scenario, and different camera viewpoints. In addition, various characters and backgrounds used for creating SynPlay are also visible.

# E  BROADER IMPACT AND LIMITATIONS

**Broader impact.** Utilizing real human datasets frequently entails inherent privacy concerns. We hope that our endeavors to enhance synthetic human data, moving it one step closer to real-world fidelity, will contribute to alleviating these challenges.

**Limitations.** SynPlay was developed to provide richer representations of human appearance for tasks that involve localizing human in the scenes. We recognize the significance of incorporating distinctive features from diverse object categories. For future work, we are aiming to expand the SynPlay to encompass a wider array of categories, thereby enriching its training capabilities.

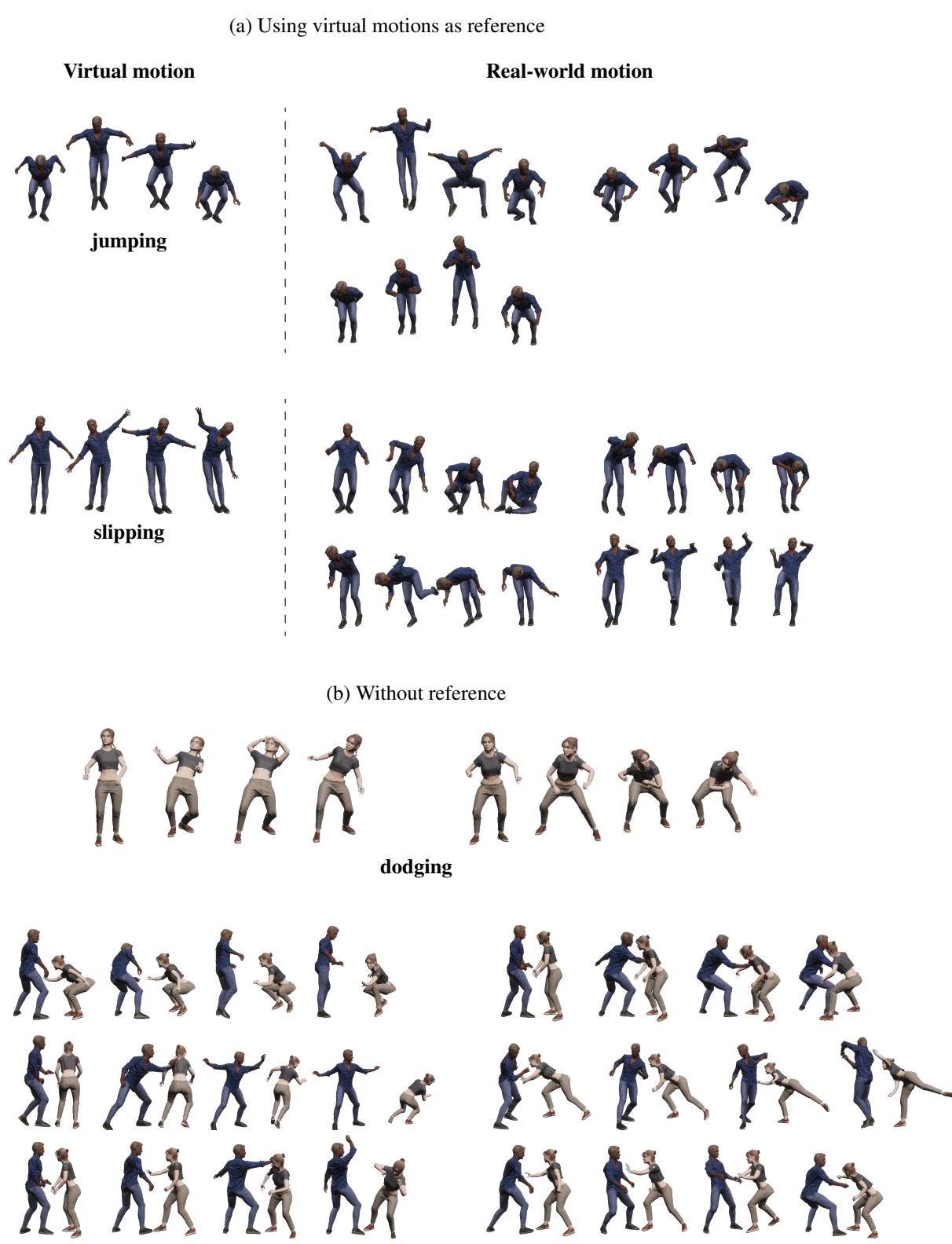

Figure 11: **Real-world motion examples.** Real-world motions are acquired either (a) by mimicking reference motions or (b) by exhibiting potential in-game motions without any reference that align with the given game rules. Wearable motion scanners are used for all the cases. Note that the semantic labels (e.g., jumping, dodging) were not provided at the time of capture; they are included in the figure only for the convenience of the readers.

(1) red light, green light

| | | |
|---|---|---|
| UGV | UAV med-alt | UAV low-alt |
| CCTV back | UAV high-alt | CCTV side |

(2) sugar candy

| | | |
|---|---|---|
| CCTV side | UAV high-alt | UGV |
| UAV low-alt | CCTV front | CCTV back |

(3) tug-of-war

| | | |
|---|---|---|
| UAV low-alt | CCTV front | UAV high-alt |
| CCTV back | UAV low-alt | UGV |

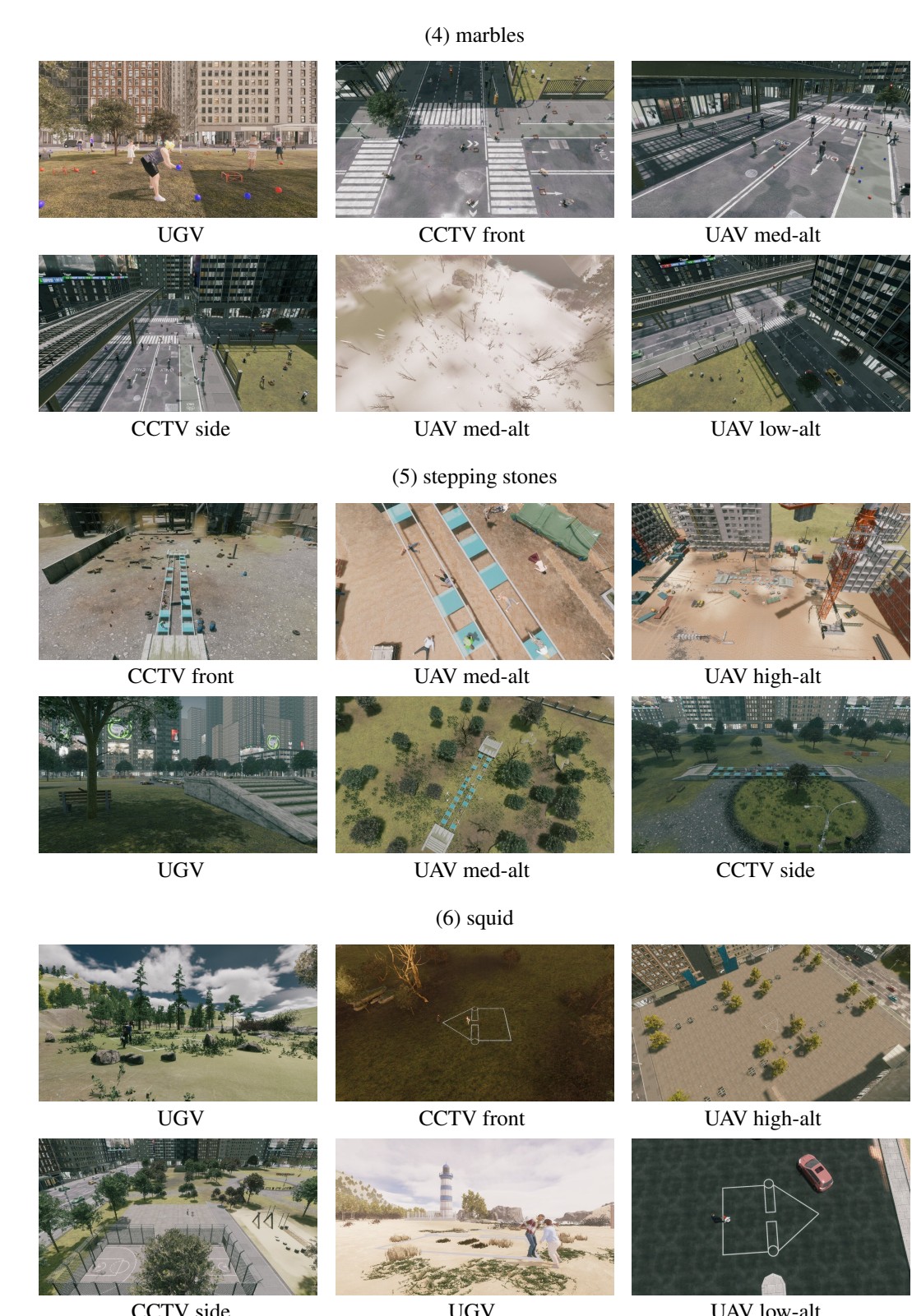

Figure 12: **More example images from SynPlay** are shown for all six Korean traditional games, each with various camera viewpoints.

