# OpenReview forum: "SynPlay: Importing Real-world Diversity for a Synthetic Human Dataset"
_ICLR.cc/2025/Conference — ICLR 2025 Conference Withdrawn Submission_

### Official Review · Reviewer_21iw · 2024-11-02

**Soundness:** 3
**Presentation:** 3
**Contribution:** 3
**Rating:** 6
**Confidence:** 3

**Summary:**

The paper introduces Synthetic Playground (SynPlay), a synthetic human dataset designed to showcase the diversity of human appearance. It emphasizes realistic human motions and multiple camera viewpoints to achieve this diversity. SynPlay captures extensive visual attributes from various angles, comprising over 73,000 images and 6.5 million human instances, and improves model training accuracy for human detection and segmentation, particularly in data-scarce scenarios.

**Strengths:**

1. The paper is well-written and easy to follow;
2. The experiment section provides detailed results and in-depth analysis;
3. The supplementary materials help with understanding the paper.

**Weaknesses:**

1. Necessary preliminary information about the "Squid Game" is missing. For example, how does the author extract the predefined motions and virtual scenarios for Unity rendering?
2. Does the dataset have copyright concerns?
3. Why do most datasets not enjoy a performance gain as shown in Figure 4?
4. The FID metric compares the distribution between two image datasets and is common for generative tasks. It's confusing to use FID to measure the diversity in Table 5.

**Questions:**

See above.

---

### Official Review · Reviewer_pswQ · 2024-11-04

**Soundness:** 3
**Presentation:** 3
**Contribution:** 2
**Rating:** 5
**Confidence:** 4

**Summary:**

This submission introduces a new synthetic human dataset. The authors claim that the dataset is new and important because of that it uses realistic human motions and poses, as well as that it renders multiple camera viewpoints for human instances. The details of the construction of the dataset are discussed and the authors conducted various experiments  on different human-centric computer vision tasks to demonstrate the value of the dataset including human detection, semantic segmentation, as well as few-shot and cross-domain human detection.

**Strengths:**

1) the constructed dataset has large number of frames and diverse human motion.
2) the motion design was based on rules of games, which is interesting and I can see potential in how this can become valuable to the community in applications such as embodied learning.
3) the impacts of including the proposed dataset in training models targeting various human-centric tasks were quantitatively evaluated and discussed with quite some useful insights

**Weaknesses:**

1) while the paper emphasizes on realistic human motion, realistic appearance of the human instances (the outfits, skin, and the environment/background) as well as background in the dataset seems to be rather unrealistic, which may cause large domain gap.
2) the two main contributions claimed for the proposed dataset are a) using realistic motion, b) multi-view are both not new (e.g. SURREAL, CMU Panoptic, PKU-DyMVHumans etc.). While there may be differences in size or motion diversity, they seem incremental.
3) while the motion diversity / sampling space seem to be larger, the pre-defined games and scenarios are somewhat limiting the actual motion diversity, which was not discussed/evaluated in the submission. It would be helpful to visualize the distributions and diversity of motion patterns assigned to the simulated human instances.
4) the dataset was generated/rendered at a frame rate of 1fps, which may defeat the purpose of using realistic motion priors to generate motion patterns for the dataset as realistic motion patterns make most differences in small details - sampling at such a low frame rate would lose all those details. In this case it is unclear how much differences can there be between the initial motion patterns and the realistic motion patterns.
5)  performance gains on MSCOCO and few-shot cross domain human detection seem incremental, given its larger data size.
6) lack of ablation study on how important using mocap motion data is w.r.t. downstream task performances when training with the proposed dataset. E.g. the authors could consider evaluating performance of downstream tasks training with the data generated with and without the realistic mocap motion patterns.

**Questions:**

the reviewer would like the authors to clarify:
1) whether the authors plan to opensource the data generation pipeline to alleviate the limited games and scenarios
2) why is the mocap motion prior still important given such a slow rendering frame rate of 1fps
3) impact of the domain gaps caused by overall unrealistic appearance
4) were the motion patterns also assigned according to age/gender groups? i.e. only motion patterns captured from children were applied on children characters?

---

### Official Review · Reviewer_7Cn4 · 2024-11-04

**Soundness:** 2
**Presentation:** 2
**Contribution:** 2
**Rating:** 3
**Confidence:** 4

**Summary:**

The paper introduces a new synthetic human dataset named SynPlay, which enhances the diversity of human appearances by diversifying the motions performed and the viewpoints captured. To diversify the motion types, the authors propose a rule-guided motion design that provides high-level, less-detailed guidance to control the human characters. To diversify the viewpoints, the dataset was captured using three widely used camera views: UAV, UGV, and CCTV. SynPlay demonstrates improved performance compared to training from scratch or using other synthetic datasets in tasks related to human detection and segmentation.

**Strengths:**

It is highly appreciated that the authors focused on diversifying two important factors—human motions and viewpoints—in synthetic human datasets. The experimental results demonstrated that the proposed dataset, with its greater diversity in these two factors, can improve model performance compared to training from scratch or using other synthetic datasets.

**Weaknesses:**

1. Novelty: The focus on human motions and viewpoints in the proposed dataset is promising. However, the methods to generate a dataset with more diverse motion types and viewpoints seem to have limited novelty. Specifically, the dataset heavily relies on game rules from six traditional VR games, as well as viewpoints provided by VR tools. This approach appears to be more about better leveraging existing tools rather than proposing new tools for generating improved datasets. Additionally, Figure 1 reveals that several human figures are of low quality, with some displaying unnatural poses, such as the individual in the last image.

2. Clarity of Writing: The writing could be clearer, as some sentences are difficult to follow. For instance:
- Line 212-213: "Note that, while a uniquely designed scenario is used for a unique sequence, the same motion evolution graph is used for all the sequences captured under the same game rule." Could you further elaborate on what this means?
- Experiment Settings (Tables 1 and 2): It would be beneficial to provide more details about the experiment settings. For example, in Table 2, are all three rows trained with the same number of iterations? The term "real data" is ambiguous too. Could you clarify what is "real" data in Tables 1 and 2?
- Line 362: There is no analysis explaining why "The anticipated synergistic effect appears in all cases except in one case."
- Correspondence between Section 4.1 and Table Results: The correspondance between the narrative in Section 4.1 and the results in the tables is unclear. For instance, the statement "Moreover, using SynPlay only provides comparable accuracy to using MS COCO when used indirectly through fine-tuning on the real dataset" is confusing. Could you specify which row in Table 3 this refers to?

**Questions:**

Please clarify my questions in Weaknesses.

---

### Note · Authors · 2024-11-15

I have read and agree with the venue's withdrawal policy on behalf of myself and my co-authors.